# Research on Ecological Infrastructure from 1990 to 2018: A Bibliometric Analysis

**Shoukai Sun [1,2], Yuantong Jiang [1,2]**  **and Shuanning Zheng [1,*]**

1   Key Laboratory of Urban Environment and Health, Institute of Urban Environment, Chinese Academy of Sciences, Xiamen 361021, China; sksun@iue.ac.cn (S.S.); ytjiang@iue.ac.cn (Y.J.)
2   University of Chinese Academy of Sciences, Beijing 100049, China
*   Correspondence: snzheng@iue.ac.cn; Tel.: +86-592-619-0675

**Abstract:** Ecological infrastructure (EI), as the composite system on which the sustainable development of cities depends, has attracted worldwide attention. Considering refined methodologies and broad scope, researchers lacked overall understanding of research patterns and evolutionary processes on EI-related issues. In this study, we applied the bibliometric approach to describe the current situation of EI-related research, and reveal research trends. Based on the Web of Science Core Collection, the bibliometric analysis of EI-related publications from 1990 to 2018 was performed to discuss the history and present research situation of EI, and preview research prospect. The results showed that: (1) the number of EI-related publications has surged astonishingly worldwide over the last three decades; (2) countries in Europe and North America were the first to invest heavily in EI-related research, while China started later but subsequently developed rapidly; (3) the EI-related research focuses were EI-related management, methods for countering extreme meteorological phenomena, providing ecosystem services, and protecting biodiversity; and (4) the EI-related research frontiers included the design of EI, policy research on EI, role of EI in environmental governance, and research on the adaptability of EI.

**Keywords:** ecological infrastructure; bibliometric analysis; visualization analysis; co-word analysis; co-citation analysis; co-authorship analysis

## 1. Introduction

Since the first industrial revolution, rapid industrial development and urbanization have led to economic prosperity and the gradual replacement of the natural landscape with municipal infrastructure or impermeable layers. Moreover, the severe ecological or environmental problems triggered by industrialization and urbanization such as urban heat island [1–3], flood disasters [4–6], and water eutrophication [7,8] directly or indirectly threaten sustainable ecosystems and public welfare [9]. In order to tackle the severe ecological or environmental problems and build sustainable cities, it is urgent to find a way to integrate or couple urbanization and the processes of ecological and social environments [10]. Starting in the late 20th century, some developed countries, represented by the United States of America, Sweden, and the United Kingdom, gradually began to realize the potential of urban infrastructure for ecological protection and attempted to draw up plans for the integration of urban development and nature conservation. For example, Maryland carried out the Greenways Movement, Smart Growth as well as Neighborhood Conservation, and developed an evaluation system for green infrastructure using the Geographic Information System [11–13]. The United Kingdom issued "Planning Policy Guidance 2: Green Belt" in 1988 to guide urban and rural planning and coordinate the relationship between urbanization and natural resources [14–16].

Ecological infrastructure (EI), which was first introduced in 1984 in UNESCO's Man and Biosphere Programme (MAB), was one of the five principles of ecological city planning which were proposed based on the urban ecological system research reports of 14 cities around the world. The original concept of EI was described as using "natural landscape and natural areas as the framework for spatially organizing the city", and emphasized the sustainable support of natural landscapes and hinterland to cities [17,18]. The prototypes of EI can be traced back to urban parks in the 1850s, such as Birkenhead Park in Liverpool, England and the "Emerald Necklace" project in Boston, USA, etc. These projects attempted the planning and design of urban green open space to provide recreation and aesthetics services for visitors and to improve the public environment, but there was no scientific and systematic theoretical guidance during this period. By 1924, the International Conference on Urban Planning was held in Amsterdam, and the concept of the "satellite city" was proposed, which provided feasible solutions to the problem of urban sprawl through the construction of greenbelts [19]. After 1960, on the basis of landscape ecology, island biogeography theory and metapopulation theory, the concepts such as ecological corridors and ecological networks which suggest to connect isolated habitat patches to help reduce the negative impact of habitat fragmentation on species survival appeared successively [20]. Ecologists and biologists draw from the concept of EI to solve the problems of natural landscape fragmentation and habitat loss [21,22]. So far, EI has developed into the complex with multiple functions such as protecting natural resources and biodiversity; enhancing the quantity, quality, and connectivity of green spaces inside and outside the city; promoting the healthy development of humans both physically and mentally; and improving urban living, working, and entertainment environments [23]. At the end of the 20th century, the concepts of ecological footprint and ecosystem services originated from ecological economics, providing methods for analyzing quantitatively the supply–demand relationship of regional resources and service functions [24]. These ideas further broadened people's understanding of the relationship between city and eco-environment, and provided a clear and comprehensive ideological basis for the connotation and functions of EI. Sound urban infrastructure systems can maintain the integrity of ecosystem services and ensure the healthy operation of the urban complex ecosystem [25,26]. For the foreseeable future, based on different social, economic, and environmental perspectives, scientists will continue to ponder the impact of EI on various aspects of urban composite systems to varying degrees.

Since the concept of EI was proposed, EI has been widely used to solve urban ecological or environmental problems. As the pioneers of urbanization, the Occident first introduced the concept of EI in urban planning. In 1999, "Towards a sustainable America, advancing prosperity, opportunity and a healthy environment for the 21st century", written by the President's Council on Sustainable Development, regarded EI as the strategic approach to achieving efficient and intensified urban land use and improving the environmental capacity to support humans' well-being [27]. The construction of EI in the UK was more focused on the issues of ecological protection, climate change, and the transformation of old urban areas [28]. Canadian EI concentrated on the ecological renovation of municipal infrastructure, that is, introducing ecological concepts into the construction and renovation of road engineering, municipal drainage, urban pipelines, and waste collection and treatment systems. While giving full play to the service functions, the projects should maximize the protection of natural resources and eco-environment, and economize on engineering consumables. There are different, but not contradictory, understandings of EI-related connotations affected by the ecological or environmental problems; the land use characteristics; and the laws, regulations, and socio-cultural backgrounds of different countries and regions. At present, artificial infrastructure is intertwined with the natural environment; EI needs to compensate for the ecological damage and degradation caused by engineering infrastructures and balance the multiple ecological functions required for urban development. By summarizing and referring to the understanding of EI in different countries, we defined EI as follows: functionally, EI provides comprehensive ecosystem services to maintain the sustainable development of society; spatially, EI, as the cross-scale, multi-level, interconnected ecological space, is the basic spatial framework for the integrity of ecological processes and the

protection of regional natural landscapes; infrastructure-friendly, EI is the national and regional life-support system, which aims to use eco-engineering and eco-techniques to conduct the construction and reformation of municipal infrastructure.

Well-written reviews help entry-level researchers understand the development of academic subjects. Bibliometrics is a burgeoning interdiscipline which involves the quantitative analysis of knowledge carriers using mathematical and statistical methods [29,30]. Modern information technology directly promotes the development of bibliometrics; many bibliometric pieces of software, such as CiteSpace, BibExcel, VOSviewer, and HistCite, provide different requirements based on user preferences. Bibliometric analysis has been widely used across different disciplines to understand the development processes in the specific fields, such as the time distribution of academic achievements, scientific cooperation, research contents, and the publication distribution of countries and institutions [31–37]. Bibliometric analysis has been used in many fields, including physical sciences, social sciences, and medical sciences [38–42]. An in-depth understanding of the current situation and future tendencies of EI-related research will help researchers define imminent issues. This work will assist policymakers in formulating effective and scientific construction strategies. This study analyzed EI-related literature published between 1990 and 2018, obtained from the Web of Science Core Collection, and carried out a bibliometric analysis of the retrieved literature using CiteSpace software, an information visualization software for dynamic network analysis. This paper aimed to solve the following questions:

1. What are major journals and their annual distribution characteristics in the field of EI?
2. What are the distribution characteristics of publication activities by countries and institutions in the field of EI?
3. What are research hotspots and frontiers in the field of EI?

## 2. Research Methods and Data Sources

### 2.1. Research Methods

This paper used data mining, measurement science, text analysis, and other means to carry out visualization analysis and mapped knowledge domains of EI-related research. Serialized knowledge domains reveal many implicit and complex relationships among knowledge elements or knowledge clusters, including interaction, intersection, evolution, and derivation, which provide a concentrated presentation of the evolution process of scientific knowledge [43].

In this study, CiteSpace, a software developed by Dr. Chaomei Chen from Drexel University, was chosen to map knowledge domains of EI-related literature published between 1990 and 2018. CiteSpace has become a common tool for text analysis, data mining and visualization analysis [38]. CiteSpace software can map three types of scientific knowledge domains for citing publications and cited publications—namely co-citation network, collaboration network, and co-word network—and intuitively express the knowledge frameworks and evolution processes of research fields. The co-citation network is a process of reorganizing dissociated knowledge elements and revealing knowledge frameworks. The co-citation process can be regarded as a filtering mechanism for peer researchers to jointly evaluate and screen research results. The identification of frequently cited literature can in turn help with an understanding of research focuses and intellectual bases [44,45]. Meanwhile, the collaboration network identifies the distribution of major research groups in specific fields by displaying the cooperative relationships among different authors, institutions, countries, and regions. The co-word network shows research hotspots and the evolution tendency of topics by analyzing the frequency of keywords and relationships between them.

Furthermore, considering the enormous amount of data, in order to avoid the confusion of scientific knowledge domains due to overly complicated and dense relationships, this study selected the "Pathfinder" network-pruning algorithm provided by CiteSpace software and assisted "Pruning the Merged network" to improve the clarity of the resultant network visualization. The "Pathfinder"

network-pruning algorithm selects significant relationships among neighboring networks according to the principle of triangle inequality to reduce link crossings and remain nodes [46].

*2.2. Data Sources*

EI-related scientific publications were retrieved from the Web of Science Core Collection, with time restriction from 1990 to 2018. In the process of document information retrieval, in order to obtain a satisfactory retrieval result, it is crucial to set up the retrieval formula that can reveal information requirements comprehensively and correctly. "Ecological Infrastructure" was used as the search term to collect total publications with the phrase in their titles, abstracts, or keywords. The search term was not expanded to make the retrieval result more specific and accurate, but may not have been comprehensive enough. The full record and cited references of 2599 original publications were downloaded for further analysis. The full record of EI-related literature, including title, abstract, keywords, document type, author information, funding, etc., facilitated the in-depth bibliometric analysis. Of all eight document types, "article" (1810) was the most frequent document type comprising 69.64% of all publications, followed by "proceedings paper" (712; 27.40%), and "review" (185; 7.12%) (Figure 1). As "article" and "proceedings paper" represented the large majority of total publications in EI-related research, two document types were selected for further analysis with deliberate exclusion of others, with a total of 2392 publications.

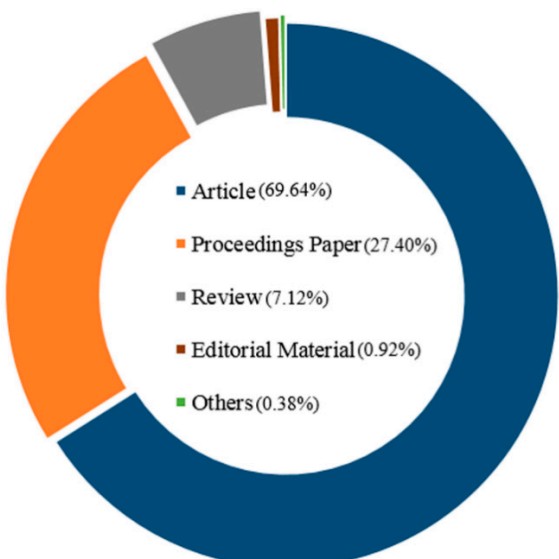

**Figure 1.** Document types and the proportion of each type of EI-related research.

## 3. Results and Discussion

*3.1. Temporal Distribution of Publications*

Of the 2392 EI-related publications that were retrieved from the Web of Science Core Collection (see Section 2.2), 2385 were selected for further analysis by duplicate checking and filtering using the "Remove duplicates (WoS)" function of CiteSpace. Figure 2 shows the annual number of EI-related publications from 1990 to 2018. After 1990, EI-related research saw a booming growth, a total of 87 papers were published in the 1990s, 451 in the 2000s, and 1847 from 2010 to 2018. The number of EI-related publications exceeded 50 for the first time in 2006 (Figure 2). The growth trajectory shows that EI-related research has become an important research topic, which is consistent with the increasing attention of the academic community on the impact of EI on urban ecosystems and society.

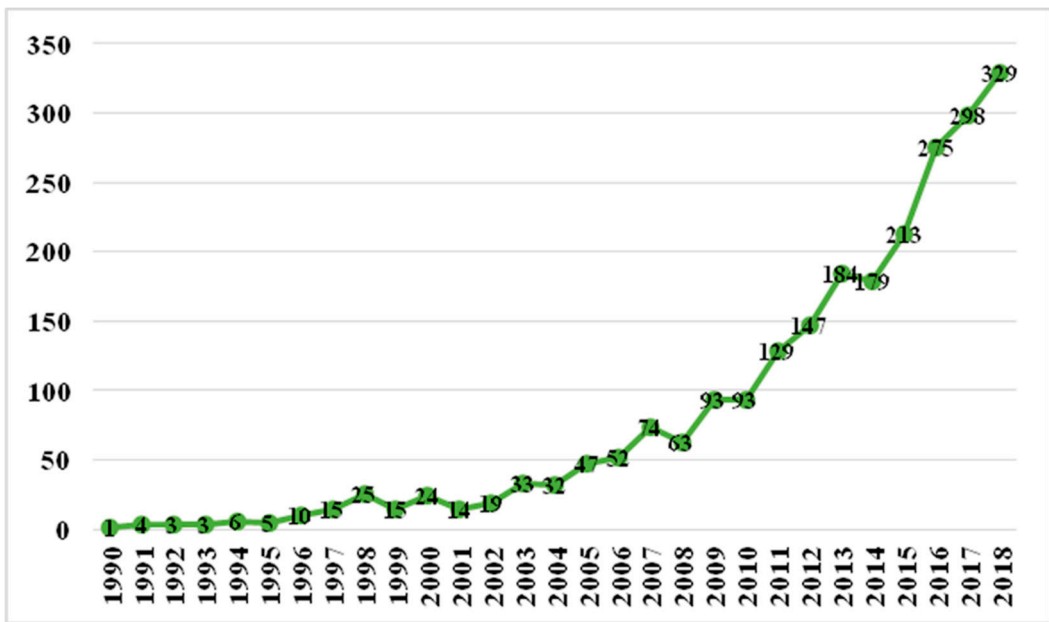

**Figure 2.** Growth trajectory of EI-related publications, 1990–2018.

### 3.2. Major Journals and Annual Distribution Characteristics

The number of journals that published EI-related literature increased significantly from 1990 to 2018, reaching 76, 268, and 617 in 1990-1999, 2000-2009 and 2010-2018, respectively. This growth phenomenon could be attributed to (1) literature written by researchers with different academic backgrounds and diversified research themes need to be published in journals with different aims and scope (Figure 3) and (2) the continuous emergence of new scientific journals.

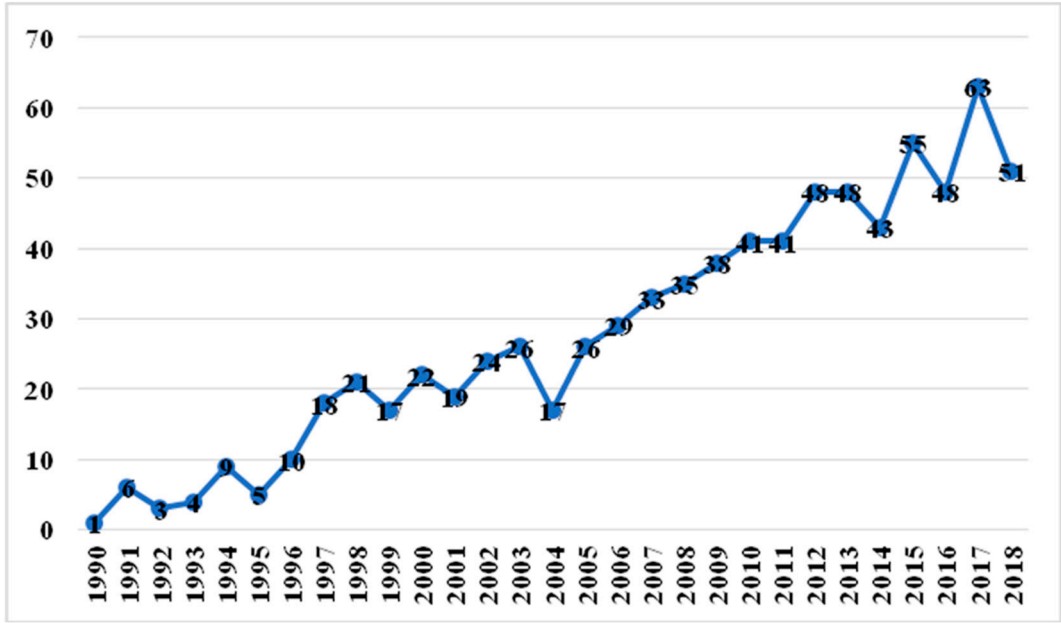

**Figure 3.** Research areas of EI-related publications, 1990–2018.

Although many journals published EI-related literature, the majority of publications were distributed in few major journals (Figure 4). From 1990 to 2018, the top 10 most productive journals were selected as the major journals in the EI-related field, with a total of 384 papers, accounting for 16.10% of total literature. "Other" journals contributed cumulatively 2001 papers, because of the lower

publication frequencies, the journals' names were not listed in detail. This activity can assist beginners in screening EI-related literature in major journals and initially determine the EI-related research areas. The percentage of the top 10 most productive journals was low, which reveals the breadth of publications distribution and broad interests of EI-related researchers. The journals, *Sustainability* (MDPI) and *Landscape and Urban Planning* (Elsevier), ranked first and second on the list of most productive journals for EI-related research from 1990 to 2018, with more than 50 publications each. Ranking journals based on the number of papers published can reveal the interests of researchers. The aim of *Sustainability* is to "meet the challenges relating to sustainability and achieve sustainable development using socio-economic, scientific and integrated approaches" (*https://www.mdpi.com/journal/sustainability/about*), while the aim of *Landscape and Urban Planning* is to "enhance the understanding of landscape concepts and applications, and coordinate social and ecological values to ensure the sustainability of the landscape" (*https://www.journals.elsevier.com/landscape-and-urban-planning*). The frequency of EI-related literature published in these two journals indicated that the coordination of the relationship between ecology and socio-economic development through EI has been a research focus.

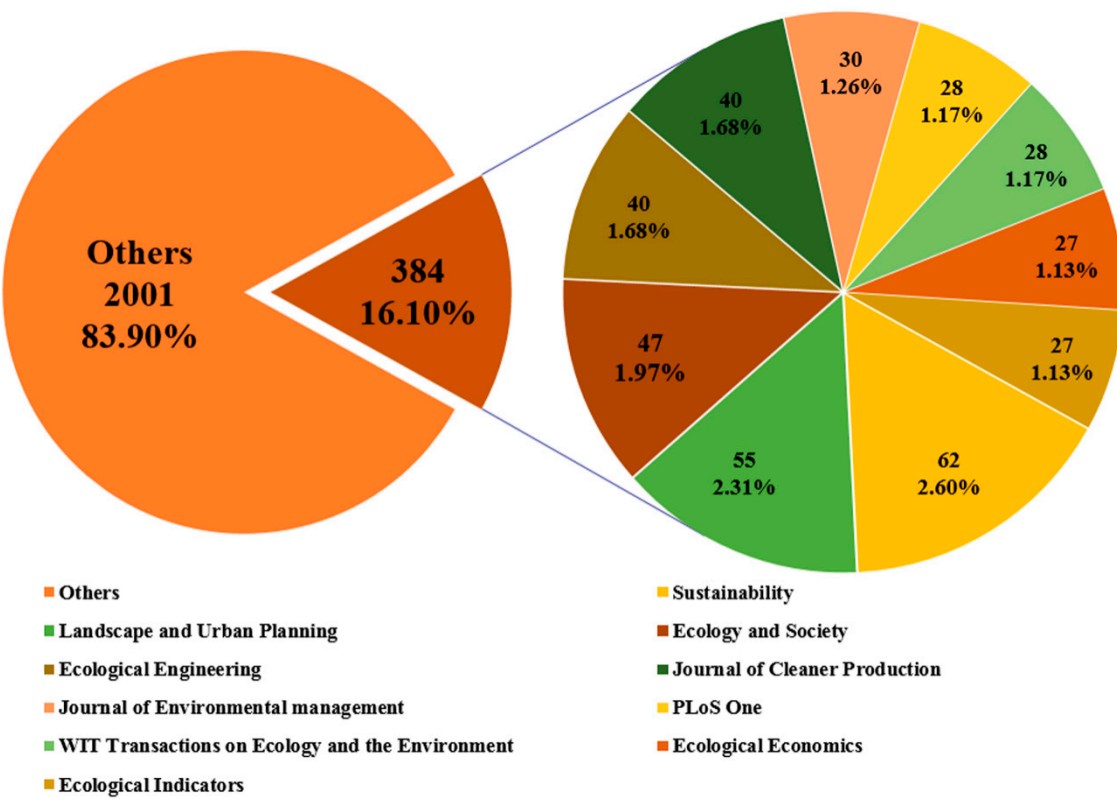

**Figure 4.** The top 10 most productive journals according to EI-related publications, 1990–2018.

Consider the difference in the number of documents published in major journals each year; Figure 5 shows the annual distribution characteristics of the top 10 most productive journals from 1990 to 2018. The initial journals to publish EI-related literature were *Economic Engineering*, *Journal of Environmental Management*, and *Ecological Economics*, which began in 1996. The journals *Landscape and Urban Planning*, *Economics and Society*, and *Journal of Cleaner Production* all began to publish EI-related literature in the early 21st century, showing an overall growth trend. Two journals, *Sustainability* and *PLoS One*, began to publish EI-related literature in the 2010s, and the number of EI-related papers published in these journals increased sharply. Obviously, the number of EI-related papers in *Sustainability* increased from 2 in 2012 to 23 in 2018, with a total of 62. In contrast, the number of papers published by *WIT Transactions on Economics and the Environment* decreased from 6 in 2005 to 1 in 2018, suggesting decreasing participation in the field of EI.

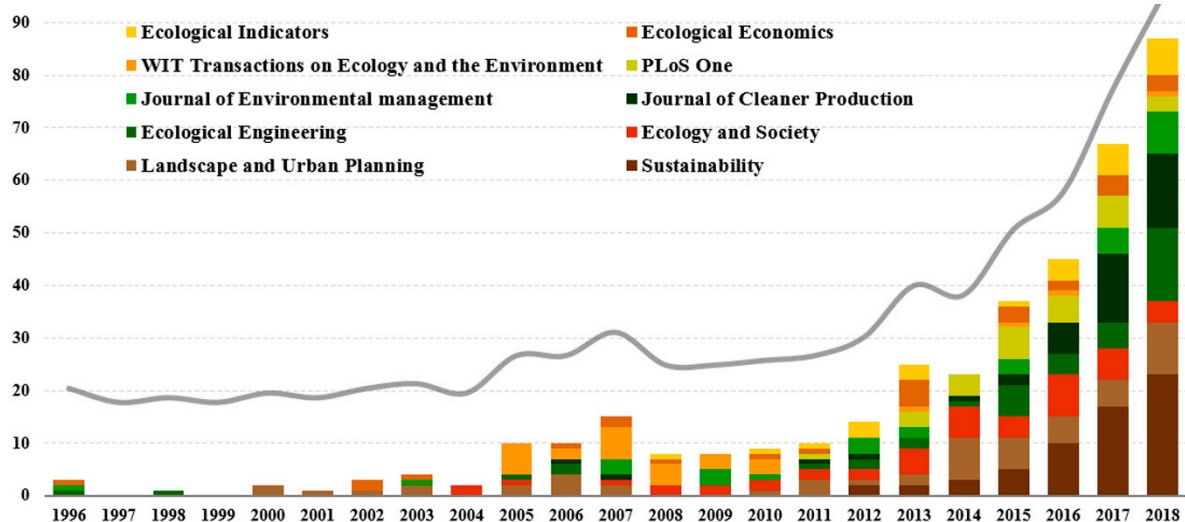

**Figure 5.** Annual distribution characteristics of the top 10 journals according to EI-related publications, 1990–2018.

*3.3. Publication Distribution of Countries and Institutions*

3.3.1. Publication Distribution of Countries

The addresses and affiliations of authors are the basis to determine the contribution of different countries and institutions. In this study, based on the analysis results of CiteSpace, the three-dimensional hotspots map was developed to represent the frequency and geographic distribution of publication activities around the world. Publications originating from England, Scotland, Northern Ireland, and Wales were classified as the UK's contributions to EI-related research. As shown in Figure 6, countries were divided into ten grades according to the number of EI-related papers they published, and the height represents betweenness centrality that can measure the importance of countries in cooperative relationships; countries with high betweenness centrality can be regarded as important hubs to connect multi-country research [47]. The result showed that EI-related research was dominated by western countries and had attracted global attention. From 1990 to 2018, the USA, China, Australia, and the UK published more than 200 pieces of literature each (Table 1), and made outstanding contributions to EI-related research, with a total of 1421 publications, followed distantly by other countries. The participation of developing countries in EI-related research is still relatively low. According to the analysis results from CiteSpace, New Zealand, Denmark, South Africa, France, Belgium, the UK, Japan, and Switzerland had high betweenness centrality and were important communication hubs in the cooperation network (Table 1). At present, EI-related research has become a critical issue around the world, and the globalization trend in EI-related research is obvious.

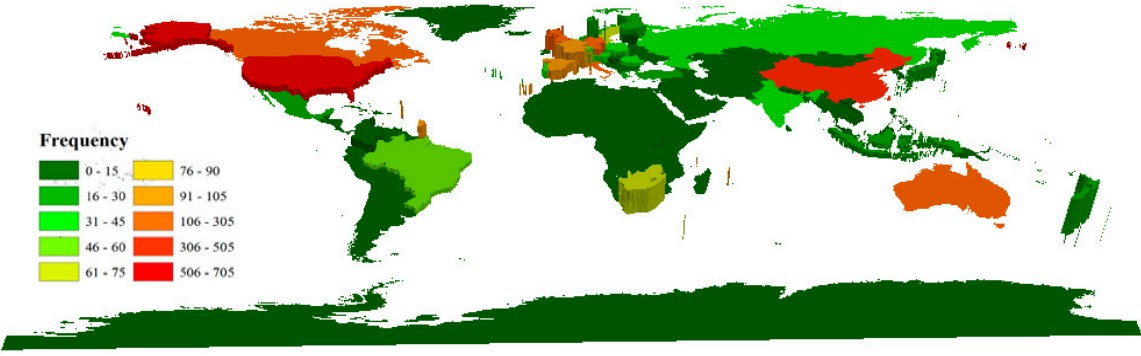

**Figure 6.** Global distribution according to EI-related publications, 1990–2018.

**Table 1.** The top 25 most productive countries in the EI-related field, 1990–2018.

| No. | Country | Frequency | Centrality | Year |
|-----|---------|-----------|------------|------|
| 1 | USA | 629 | 0.18 | 1996 |
| 2 | PEOPLES R CHINA | 344 | 0 | 1998 |
| 3 | AUSTRALIA | 229 | 0 | 2003 |
| 4 | United Kingdom | 219 | 0.44 | 1999 |
| 5 | GERMANY | 161 | 0.37 | 1998 |
| 6 | CANADA | 130 | 0 | 1998 |
| 7 | NETHERLANDS | 113 | 0.06 | 1994 |
| 8 | ITALY | 113 | 0 | 2007 |
| 9 | SPAIN | 94 | 0.29 | 2003 |
| 10 | FRANCE | 92 | 0.51 | 2003 |
| 11 | SWEDEN | 66 | 0 | 2004 |
| 12 | SOUTH AFRICA | 62 | 0.71 | 2009 |
| 13 | SWITZERLAND | 55 | 0.44 | 2006 |
| 14 | BRAZIL | 48 | 0.12 | 2009 |
| 15 | POLAND | 44 | 0.18 | 2008 |
| 16 | INDIA | 41 | 0 | 2004 |
| 17 | RUSSIA | 41 | 0 | 2010 |
| 18 | AUSTRIA | 36 | 0.18 | 2010 |
| 19 | ROMANIA | 35 | 0.12 | 2002 |
| 20 | PORTUGAL | 32 | 0.23 | 2009 |
| 21 | JAPAN | 29 | 0.4 | 2004 |
| 22 | FINLAND | 28 | 0.33 | 2005 |
| 23 | DENMARK | 27 | 0.85 | 2007 |
| 24 | BELGIUM | 27 | 0.45 | 2012 |
| 25 | NORWAY | 25 | 0 | 2013 |

### 3.3.2. Publication Distribution of Institutions

The contribution of institutions was evaluated by the affiliation of authors; institutions with frequent publication activities were considered to have tremendous strength in EI-related research. Table 2 listed the top 10 most productive institutions that participated in EI-related research in the three periods of 1990–1999, 2000–2009, and 2010–2018. The results show the increasing number of institutions participating in EI-related research from 1990 to 2018. In addition, based on results obtained by CiteSpace software, we developed projection maps to reveal the geographical and temporal distribution of institutions and publication frequencies of institutions.

**Table 2.** The top 10 most productive institutions in EI-related research during 1990–1999, 2000–2009, and 2010–2018.

| 1990–1999 | | | 2000–2009 | | | 2010–2018 | | |
|-----------|---|-----|-----------|---|-----|-----------|---|-----|
| Institutions | N | AC | Institutions | N | AC | Institutions | N | AC |
| Leiden University | 3 | 1.39 | Chinese Academy of Sciences | 13 | 2.98 | Chinese Academy of Sciences | 60 | 4.62 |
| Wageningen University Research | 3 | 0.6 | University of California System | 13 | 3.84 | University of California System | 52 | 8.84 |
| DLO | 2 | 0.48 | Commonwealth Scientific Industrial Research Organisation | 11 | 6.62 | United States Department of Agriculture | 38 | 6.81 |
| North Carolina State University | 2 | 0.43 | United States Department of Agriculture | 11 | 5.05 | United States Department of the Interior | 37 | 8.1 |
| Radboud University Nijmegen | 2 | 0.07 | Helmholtz Association | 8 | 4.37 | Centre National de la Recherche Scientifique | 33 | 5.37 |
| University of North Carolina | 2 | 0.43 | State University System of Florida | 8 | 5.08 | Arizona State University | 32 | 6.91 |
| University of Twente | 2 | 2.9 | United States Department of the Interior | 8 | 6.14 | State University System of Florida | 32 | 7.09 |

**Table 2.** *Cont.*

| 1990–1999 | | | 2000–2009 | | | 2010–2018 | | |
| --- | --- | --- | --- | --- | --- | --- | --- | --- |
| AGAFE | 1 | 0.15 | Delft University of Technology | 7 | 0.73 | United States Forest Service | 32 | 7.59 |
| Alcan Deutschland GmbH | 1 | 0.16 | United States Forest Service | 7 | 7.31 | United States Geological Survey | 29 | 9.57 |
| Arthur D. Little | 1 | 1.33 | United States Geological Survey | 6 | 6.85 | Helmholtz Association | 26 | 10.46 |

Note: N—number of publications; AC—average number of citations per year.

From 1990 to 1999, most EI-related research was conducted by institutions located in the USA and Western Europe, with no more than 5 papers published by each institution (Figure 7). The most productive institutions during this period were Leiden University and Wageningen University & Research, both in the Netherlands. In the 2000s, both the number of institutions engaging in EI-related research and the number of published papers increased rapidly. Although there were several EI-related institutions in East Asia, institutions in Europe and the USA still occupied a dominant position. The European institutions which produced the most EI-related literature from 2000 to 2009 included Helmholtz Environmental Research Center, Wageningen University & Research, and Delft University of Technology. The EI-related research community in the USA was composed of government agencies represented by the U.S. Department of Agriculture, the U.S. Department of the Interior, and the U.S. Geological Survey, and higher education institutions such as the University of California system and Florida State University system (Figure 8). Meanwhile, the Chinese Academy of Sciences and the Commonwealth Scientific and Industrial Research Organization have developed into important institutions on EI. From 2010 to 2018, EI-related research activities expanded to six continents, with most of the output coming from European and American institutions such as the University of California system, Florida State University system, U.S. Department of Agriculture, U.S. Department of the Interior, U.S. National Forest Service, French National Centre for Scientific Research, Swedish University of Agricultural Sciences, University of London, and Stockholm University. However, in Africa and South America, there was low publication activity with few institutions participating in EI-related research. It is worth noting that China and Australia made remarkable progress in EI-related research in the past decade. In China, the Chinese Academy of Sciences, Tsinghua University, Peking University, and Beijing Normal University established EI-related research teams. The Australian research community mainly consisted of the Commonwealth Scientific and Industrial Research Organization, University of Queensland, University of Melbourne, and Australian National University (Figure 9).

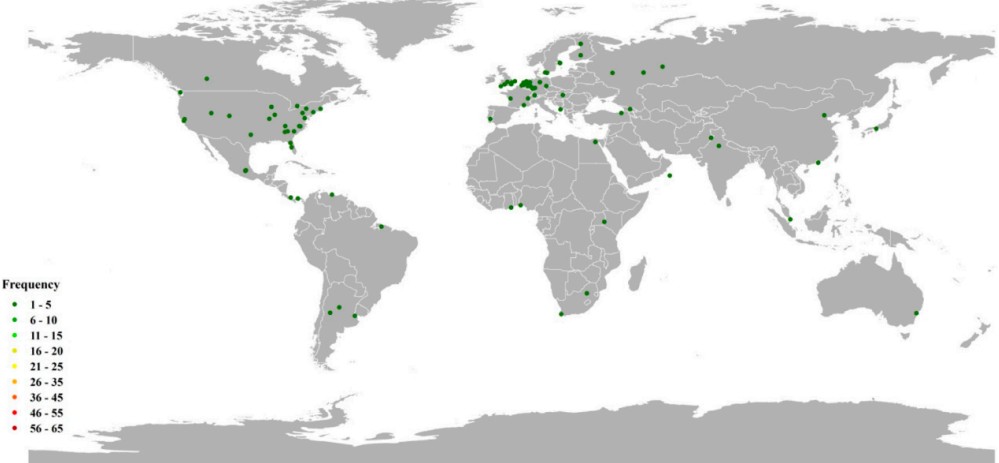

**Figure 7.** Geographical distribution of EI institutions, 1990–1999.

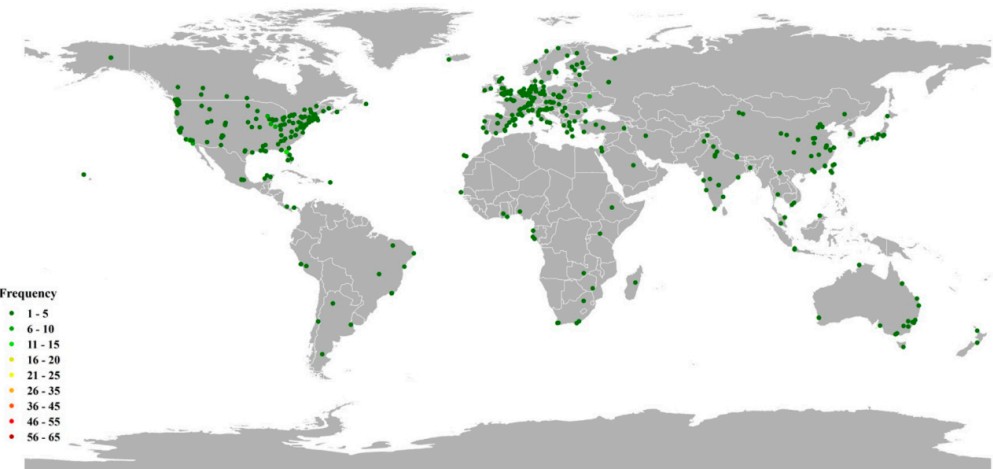

**Figure 8.** Geographical distribution of EI institutions, 2000–2009.

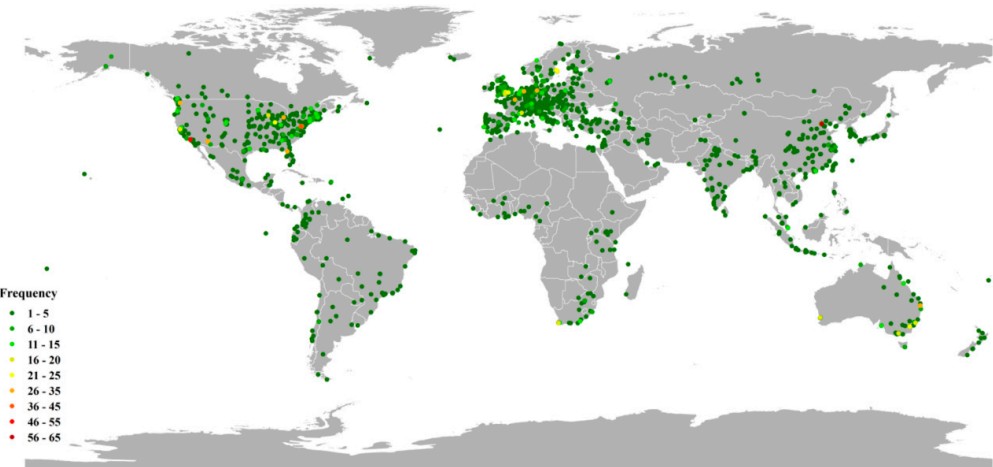

**Figure 9.** Geographical distribution of EI institutions, 2010–2018.

From 1990 to 2018, the USA always maintained a leading position in EI-related research and publishing the most literature. In 2000–2009 and 2010–2018, in terms of the output of EI-related literature, the institutions placing sixth and seventh on the list of 10 most productive institutions were located in the USA (Table 2). Table 2 shows the annual citation frequency of papers of various institutions that reflect its academic influence. Although the Chinese Academy of Sciences was the most productive institution from 2000 to 2018, its annual citation frequency remained low.

*3.4. Highly Cited Publications*

Table 3 lists the most cited publications in the field of EI research from 1990 to 2018, containing additional information: author, title, journal, country/institution, total citations (as of December 10, 2019). The most frequently cited paper, with 658 citations, is "Urban green space, public health, and environmental justice: The challenge of making cities 'just green enough'", authored by Jennifer R. Wolch et al., and published in *Landscape and Urban Planning* in 2014. This paper regarded the accessibility of urban green space as an issue of "environmental justice", and reviewed research on the relationships between urban green space and public health, emphasizing the importance of urban green space in improving public health. Their study found the accessibility of urban green space was improved by promoting small-scale and scattered ecological nodes [48]. Many subsequent studies of the accessibility of urban green space and environmental justice cited the results of Wolch et al.

and discovered the impacts of the geographical and temporal characteristics of urban green space on residents' health and well-being, which have become the main topic of EI research in recent years.

**Table 3.** The top 15 cited publications in EI-related field, 1990–2018.

| No. | Author | Title | Journal | Country/Institution | Citations |
|---|---|---|---|---|---|
| 1 | Wolch, JR et al. (2014) | Urban green space, public health, and environmental justice: The challenge of making cities 'just green enough' | LANDSCAPE URBAN PLAN | USA/Univ Calif Berkeley and Univ Michigan; Australia/Griffith Univ | 658 |
| 2 | Anderies, JM et al. (2004) | A framework to analyze the robustness of social-ecological systems from an institutional perspective | ECOL SOC | USA/Arizona State Univ and Indiana Univ, Bloomington | 649 |
| 3 | Gleick, PH (2003) | Global freshwater resources: Soft-path solutions for the 21st century | SCIENCE | USA/Pacific Inst Studies Dev Environm & Secur, Oakland | 590 |
| 4 | Meehl, GA et al. (2000) | An introduction to trends in extreme weather and climate events: Observations, socioeconomic impacts, terrestrial ecological impacts, and model projections | B AM METEOROL SOC | USA/Natl Ctr Atmosfer Res, Climate & Global Dynam Div and Natl Climate Data Ctr | 470 |
| 5 | Wulder, MA et al. (2012) | Opening the archive: How free data has enabled the science and monitoring promise of Landsat | REMOTE SENS ENVIRON | Canada/Nat Resources Canada, Canadian Forest Serv, Pacific Forestry Ctr; USA/NASA, Biospher Sci Lab, Goddard Space Flight Ctr | 456 |
| 6 | Gomez-Baggethun, E and Barton, DN (2013) | Classifying and valuing ecosystem services for urban planning | ECOL ECON | Spain/Univ Autonoma Barcelona and Autonomous Univ Madrid | 448 |
| 7 | Bowman, DMJS et al. (2011) | The human dimension of fire regimes on Earth | J BIOGEOGR | Australia/Univ Tasmania; USA/Univ Calif Santa Barbara | 372 |
| 8 | Bolla, R et al. (2011) | Energy efficiency in the future Internet: A survey of existing approaches and trends in energy-aware fixed network infrastructures | IEEE COMMUN SURV AND TUT | Italy/Univ Genoa and Telecom Italia | 360 |
| 9 | Syphard, AD et al. (2007) | Human influence on California fire regimes | ECOL APPL | USA/Univ Wisconsin and US Geol Survey, Western Ecol Res Ctr | 297 |
| 10 | Cumming, GS et al. (2005) | An exploratory framework for the empirical measurement of resilience | ECOSYSTEMS | South Africa/Univ Cape Town; USA/Univ Florida | 250 |
| 11 | Bernhardt, ES and Palmer, MA (2007) | Restoring streams in an urbanizing world | FRESHWATER BIOL | USA/Duke Univ and Univ Maryland | 229 |
| 12 | Jackson, LE (2003) | The relationship of urban design to human health and condition | LANDSCAPE URBAN PLAN | USA/US EPA, Natl Hlth & Environm Effects Res Lab | 225 |
| 13 | Pahl-Wostl, C et al. (2007) | Managing change toward adaptive water management through social learning | ECOL SOC | Germany/Univ Osnabruck; Austria/Int Inst Appl Syst Anal | 224 |
| 14 | Jetz, W et al. (2012) | Integrating biodiversity distribution knowledge: toward a global map of life | TRENDS ECOL EVOL | USA/Yale Univ and Univ Colorado; Canada/Calgary Zool Soc, Ctr Conservat Res | 217 |
| 15 | Drew, JA (2005) | Use of traditional ecological knowledge in marine conservation | CONSERV BIOL | USA/Boston Univ | 209 |

"A framework to analyze the robustness of social-ecological systems from an institutional perspective" and "Global freshwater resources: soft-path solutions for the 21st century", which were respectively cited 649 and 590 times, ranked second and third on the list of most highly cited publications. In order to improve the robustness of social-ecological systems, the former presented an initial framework for analyzing the internal dynamics within the components of social-ecological systems which include resources, resource users, public infrastructure providers, and public infrastructures, and the interactions among the components. This literature reveals the importance of public infrastructure to improve the robustness of social-ecological systems [49]. The latter attempts to complement large-scale centralized water conservancy projects with lower-cost community-scale infrastructures, and to improve social and individual well-being per unit water used and meet basic human and ecological needs for water. This literature exploits the potential of small-scale decentralized water infrastructures for water conservation and efficiency improvements [50]. The fourth most cited paper, entitled "An introduction to trends in extreme weather and climate events: observations, socioeconomic impacts, terrestrial ecological impacts, and model projections", by the U.S. National

Center for Atmospheric Research, had 579 citations. The results revealed extreme weather phenomena caused by natural climate fluctuations or greenhouse gas-induced warming have destructive effects on infrastructure [51]. The large number of citations of these three papers showed that research on ecological engineering construction and the role of EI in improving the resilience of social-ecological systems received much more attention.

## 3.5. Research Hotspots

By analyzing the word frequency and appearance time of keywords that can express and summarize the core contents of the literature, researchers capture research hotspots and tendencies in specific fields. Considering the keywords' plural forms, abbreviations, and other differences in expression, Table 4 collated and listed the top 25 most frequently used keywords in the 2385 EI-related publications from 1990 to 2018, including three types of keywords, namely title words, author keywords, and keywords plus. We comprehensively considered the keyword ranking based on three keyword extraction methods, and selected the most popular keywords as "management", "climate change", "ecosystem services", "biodiversity", "green infrastructure", "sustainability", "conservation", "infrastructure", "resilience", and "landscape", suggesting that EI research mainly focuses on EI management, solutions to extreme weather conditions, providing ecosystem services, protecting biodiversity, etc.

**Table 4.** The top 25 title words, author keywords, and keywords plus used in EI-related field, 1990–2018.

| Rank | Title Words | Author Keywords | Keywords Plus |
|------|-------------|-----------------|---------------|
| 1 | management | green infrastructure | management |
| 2 | climate change | ecosystem services | conservation |
| 3 | conservation | sustainability | climate change |
| 4 | biodiversity | climate change | biodiversity |
| 5 | ecosystem services | biodiversity | impact |
| 6 | sustainability | resilience | system |
| 7 | infrastructure | gi | landscape |
| 8 | impact | infrastructure | ecosystem services |
| 9 | system | urbanization | model |
| 10 | green infrastructure | social-ecological system | city |
| 11 | landscape | conservation | ecosystem |
| 12 | ecosystem | sustainable development | framework |
| 13 | resilience | ecological infrastructure | infrastructure |
| 14 | city | ecology | community |
| 15 | model | adaptation | sustainability |
| 16 | land use | urban planning | land use |
| 17 | framework | china | habitat |
| 18 | ecology | urban ecology | ecology |
| 19 | community | ecological engineering | resilience |
| 20 | habitat | environment | pattern |
| 21 | urbanization | policy | social-ecological system |
| 22 | environment | remote sensing | population |
| 23 | china | land use | diversity |
| 24 | restoration | wetland | area |
| 25 | pattern | landscape | environment |

Excluding the search term "ecological infrastructure", "management" was the most frequently used keyword, which ranked first in title words and keywords plus. This revealed that the scientific management of EI is highly valued. Considering the uncertainty of ecological effects of municipal infrastructures and the complexity of socio-political factors, the traditional state-oriented and government-oriented management patterns cannot resolve the scenario-based and complex environmental problems, thus making it challenging to maximize the benefits of EI through efficient management [52,53]. The prerequisite for managing EI is understanding the mechanisms of urban ecosystem services and construction purposes of EI, and conducting collaborative management at the landscape scale and regional scale [54,55]. The management of EI transits from single-sector scale to comprehensive multi-sector scale, and supervises the design, construction, operation, and maintenance of EI to ensure the integrity of the structure and function.

"Climate change" ranked respectively second, fourth and third in title words, author keywords, and keywords plus, which suggested that studies on the relationships between EI and climate change were increasingly conducted. Indeed, in order to achieve sustainable development, researchers have regarded EI as a mitigation and adaptation strategy for climate change, and aimed at reducing the vulnerability of urban when facing extreme weather [56–58]. In addition, the possible environmental and socio-economic burdens caused by climate change may impose on EI; researchers analyzed the relationships between climate change and urban infrastructure through the modern complexity theory [59]. Current research focuses on the impact of climate change on urban hydraulic infrastructure, such as the regulation of urban storm water through drainage systems [60–64] and the contribution of urban green infrastructure for adapting to climate change and mitigating the ecological or environmental problems [58], for example, reducing $CO_2$ emissions [65,66] and improving thermal comfort [67–70].

"Ecosystem service" ranked among the top 10 in title words, author keywords, and keywords plus. The original explanation was "ecosystem services are the conditions and processes through which natural ecosystems, and the species that make them up, sustain and fulfill human life", proposed by Gretchen C. Daily in 1997 [71]. Others defined ecosystem services as the benefits derived by cities from ecosystem functions [72], or as the direct and indirect contributions of ecosystems to human well-being [72,73]. In order to meet the needs of local beneficiaries, urban ecosystem services were regarded as indicators to measure functions of EI, and integrated into the synthetic evaluation index system of urban infrastructure [74–80]. Compared with traditional green open space, EI is more effective and more suitable for complex urban areas and provides biophysical services, such as improving urban microclimates [25,81–83] and relieving the loss of biodiversity as habitats and ecological corridors [84,85]. It also provides social and cultural services, including recreational opportunities and aesthetic enjoyment [86].

"Biodiversity" ranked fourth, fifth and fourth in title words, author keywords, and keywords plus, respectively. Habitat fragmentation and loss caused by urbanization severely impact species richness and genetic diversity, thus affecting the goods and services provided by ecosystems, which became urgent issues for policymakers and planners [87–89]. Biologists and ecologists took EI as the carrier to counter habitat fragmentation, protect biodiversity, and maintain ecological balance [90–95]. According to the research of biodiversity conservation, EI focuses on assisting and guiding animals to communicate between different habitats. The connectivity of EI is key to the normal operation of natural systems, improving the quality of the natural environment to promote interactions between organisms through establishing complete habitat networks [87,96].

"Green infrastructure" (GI) ranked respectively tenth and first in title words and author keywords, which suggested that EI-related researchers and GI-related researchers share the same academic backgrounds. The concept of GI first appeared in the 1994 Florida Land Protection Report, which emphasized the protection of natural landscapes and ecosystems. Later, the GI working group, led by the Conservation Fund and the U.S. Forest Service, considered GI "our nation's natural life-support system", and suggested the construction of interconnected green space structures through landscape design and ecological planning methods. The current GI-related research can be classified into two research fields: first, providing necessary ecosystem services for urban complexes in relation to the needs of society and humans [48,97–101]; second, establishing interconnected ecological space as urban areas or wildlife corridors [102–106].

"Conservation" ranked respectively third and second in title words and keywords plus; "sustainability" ranked sixth and third in title words and author keywords, respectively; and "resilience" ranked thirteenth and sixth in title words and author keywords, respectively. EI is not a completely new planning method, and rather represents the integration of existing planning methods with a focus on environmental elements [107–109]. The three concepts mentioned above—conservation, sustainability, and resilience—can be used not only as goals and principles for the planning and construction of EI but also as the evaluation index system for municipal infrastructure. The potential of EI should be fully

explored, in order to adapt to the challenges of climate change, the degradation of the environment, and resource shortages [110,111].

### 3.6. Research Frontiers

Keyword co-occurrence analysis directly revealed the knowledge domain of EI and did not display the characteristics of temporal distribution of keywords. The time-zone view from 1990 to 2018 illustrated the evolutionary trend of the EI-related research field (Figure 10). The keywords with the strongest citation bursts mean that the frequency of occurrence increased explosively at a certain time. Combining with keywords burst detection (Figure 11), the research frontiers in this field include design, policy, governance, and adaptation.

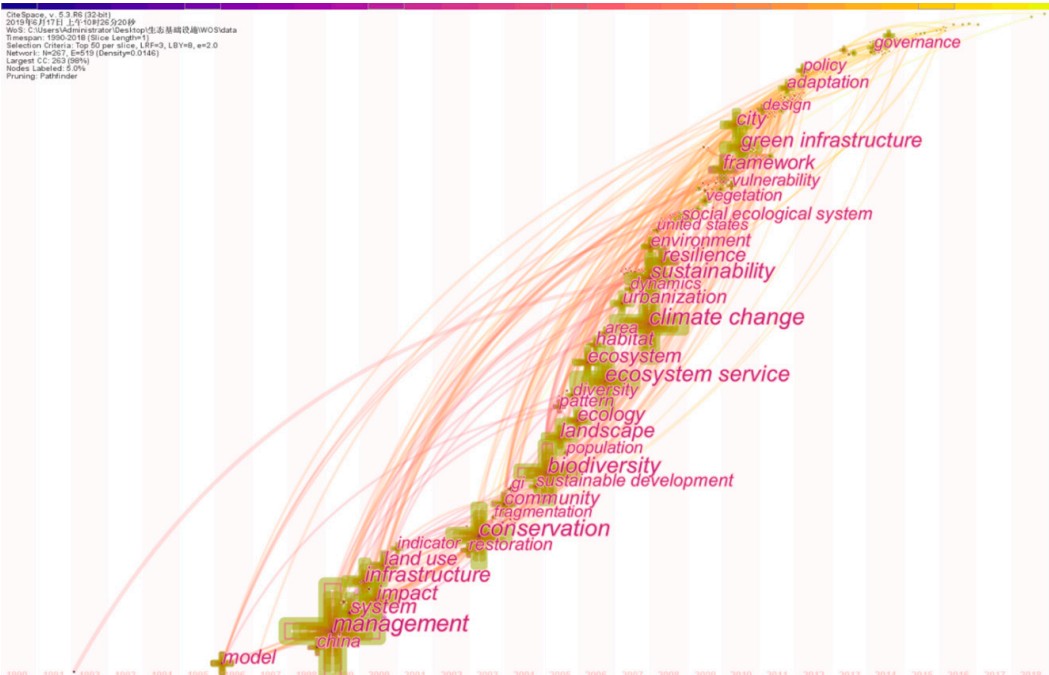

**Figure 10.** The time-zone view of keywords in EI-related field, 1990–2018.

#### Top 20 Keywords with the Strongest Citation Bursts

| Keywords | Year | Strength | Begin | End | 1990 - 2018 |
|---|---|---|---|---|---|
| system | 1990 | 3.9679 | 1999 | 2006 | |
| gi | 1990 | 10.6271 | 2003 | 2013 | |
| conservation | 1990 | 3.5863 | 2003 | 2006 | |
| sustainability | 1990 | 3.8701 | 2007 | 2009 | |
| ecological footprint | 1990 | 3.9741 | 2007 | 2012 | |
| water | 1990 | 4.1101 | 2010 | 2011 | |
| energy | 1990 | 3.8855 | 2010 | 2015 | |
| uncertainty | 1990 | 3.4568 | 2011 | 2013 | |
| adaptive capacity | 1990 | 3.6699 | 2013 | 2014 | |
| wetland | 1990 | 4.942 | 2013 | 2015 | |
| connectivity | 1990 | 6.2462 | 2013 | 2016 | |
| vulnerability | 1990 | 5.3054 | 2013 | 2015 | |
| risk | 1990 | 3.5904 | 2014 | 2016 | |
| design | 1990 | 3.4679 | 2015 | 2018 | |
| road | 1990 | 4.0046 | 2015 | 2016 | |
| knowledge | 1990 | 3.4141 | 2015 | 2016 | |
| social-ecological system | 1990 | 4.7945 | 2016 | 2018 | |
| protected area | 1990 | 4.1737 | 2016 | 2018 | |
| governance | 1990 | 3.659 | 2016 | 2018 | |
| urban | 1990 | 3.7248 | 2016 | 2018 | |

**Figure 11.** The keywords burst detection in EI-related field, 1990–2018.

### 3.6.1. Design of Ecological Infrastructure

The design of EI emphasizes the integration of ecological ideas into the design of municipal infrastructure at the initial stage of project construction, taking into account the inherent ecological losses caused by urban municipal infrastructure to ensure the integrity of structure and function of ecosystems, and provides ecosystem services needed for sustainable development [88,112,113]. Infrastructure-based ecological design is the complex concept that improves and coordinates the planning, design, and implementation of traditional infrastructure, providing a new method to deal with the cumulative impacts of municipal infrastructure on the functions and services of ecosystems, regional biodiversity, and natural resources. At the initial design stage, the needs and intentions of various stakeholders are fully considered, the goals of restoring the integrity of ecosystem structure and protecting the ecological processes and patterns are at the same priority as building technologies and socio-economic targets in the design category [113,114]. Based on multi-dimensional perspectives, EI needs to consider not only rational factors such as physical geographic conditions, architectural technologies, and construction purposes, but also pay attention to regional culture and aesthetics factors, and breaking the phenomenon of the standardization of EI-related design in order to improve construction efficiency. The aesthetic pursuit of EI is the important supplement after the ecological and socio-economic value. We suggest EI-related designs should accept and respect the existing natural context, and advocate protecting the original beauty from being distorted by anthropogenic impact. This will be a hotspot for EI-related research in the next few years.

### 3.6.2. Policy Research on Ecological Infrastructure

EI-related policy is the product of interdisciplinary research between ecologists and sociologists, which means previous research results and cases are used as prior knowledge to formulate short-term or long-term urban plans. The essence of EI-related policy is the prospective development and use of space resources; the EI-related strategy formulation requires policymakers think in terms of politics, social economy, history, geography, and technology. With the increase of scientific research on the functions of EI, it is urgent for policymakers to carry out EI-related research on non-scientific driving factors such as legislation and politics to assist in EI projects [115–117]. As the comprehensive embodiment of natural resources, culture, social policy, and other elements, the connotation of EI is far beyond simple spatial planning projects. Currently, the construction of EI is dominated by governments, and the main funding sources mainly depend on government financial appropriations, combined with social funds [13]. It is essential to promote public participation in the construction of EI to form more sustainable and scientific policies [118,119]. Because of the increasing populations and deteriorating environment, researchers consider economic assessment, comprehensive modeling, stakeholder analysis, and multi-criteria evaluation methods to formulate targeted public policies which are coordinated with the construction of sustainable EI [120,121]. EI-related policy is the multi-scale planning scheme with foresight, systematic, and integrity. The formulation of EI-related policy requires the application of ecological principles to rationally arrange ecological landscape elements, aiming at enhancing the attractiveness and competitiveness of cities.

### 3.6.3. Ecological Infrastructure Participating in Environmental Governance

This study distinguished the keyword "management" from "governance". "Management" is taken to mean "the act of running and controlling a business or similar organization" (https://www.oxfordlearnersdictionaries.com/definition/english/management?q=management), "governance" is taken to mean "the activity of governing a country or controlling a company or an organization; the way in which a country is governed or a company or institution is controlled" (https://www.oxfordlearnersdictionaries.com/definition/english/governance?q=governance). "Governance" refers to the exertion of government functions, covering laws and regulations, policies, regulations, and other rules, as well as the governments and authorities participating in environment management and

resource utilization. Environmental governance lies in the scientific application of information about ecosystem functions to the decision-making process [122,123]. Some scholars have proposed an approach to achieve the sustained health and integrity of eco-economic systems which involves a combination of governance, management, and monitoring. Governance is the process of solving trade-off problems and meeting sustainable development demand; management is the activity of implementing the vision of sustainable development, monitoring reveals the feedbacks between society and the environment, and summarizing means the observed situation and possible future trends [124]. Research has shown that, faced with conflicts in complex ecosystems, adaptive governance performs well in protecting biodiversity, providing ecosystem services, and using natural resources sustainably. At present, the research on adaptive governance focuses on the resilience of social ecosystems and environmental governance issues [125–128]. Due to environmental problems such as ecosystem degradation and the rapid consumption of natural resources, and decision-making problems such as policy fragmentation, planners cannot describe in detail the rules, behaviors, and intentions of urban environmental governance. However, the goal of urban environmental governance can be broadly understood as achieving the sustainable development of urban ecosystems. EI is a significant investment in environmental governance; guiding the construction of EI through cooperative research and participatory decision-making is effective for achieving the goal of sustainable environmental governance [129–131].

### 3.6.4. Research on the Adaptation Characteristics of Ecological Infrastructure

The Intergovernmental Panel on Climate Change (IPCC) defines "adaptation" as an "adjustment in natural or human systems in response to actual or expected climatic stimuli or their effects, which moderates harm or exploits beneficial opportunities". The in-depth understanding of adaptation is an important method to counter the uncertainty and extremes of complicated changes [132–134]. The complexity of urban areas makes it difficult to predict the independent variables, such as population and transportation, that affect the urban land scale and function layout, which makes the construction of EI passive and inflexible. EI-related planning should excavate the functions and benefits of ecological assets, focusing on coordinating the relationship between ecological protection and stakeholders such as citizens, investors and government departments. The construction of EI needs to adapt to changing urban environments and stakeholders' needs. Therefore, in the next few years, research should focus on how to solve the problems of imperfect ecological function and the imbalance of the geographical distribution of EI [135].

## 4. Conclusions

In this review, based on the Web of Science Core Collection, we conducted a bibliometric analysis of EI-related literature from 1990 to 2018 using CiteSpace software to gain clearer insight into the countries, institutions, journals, categories, highly cited publications, research hotspots, and research frontiers involved in EI research.

Results revealed that research on EI increased sharply over the past thirty years, and the largest growth in the number of EI-related papers occurred between 2000 and 2018. "Environmental sciences" and "ecology" were the most popular research categories for EI-related literature across the period of analysis. The top 10 most productive journals accounted for 16.10% of the total number of EI-related literature; the journals *Sustainability*, *Landscape and Urban Planning*, and *Ecological Engineering* published the most EI-related literature. Institutions located in the USA, China, Australia, and the UK had high productivity in total EI-related literature. Most recently, China produced the second-highest number of EI-related papers. The Chinese Academy of Sciences produced the most EI-related literature of any institution in the world, but the average of citations is low. Based on collaboration networks, Western European countries were at the center of academic cooperation in the field of EI, while China lacked significant academic influence. Additionally, keywords analysis proved to be an effective method to study hotspots and research frontiers, and its results showed that EI-related research mainly focused

on the following two aspects: first, the ecological attributes and social services of EI; and second, the application of EI in natural resource management. Therefore, the core content of EI-related research is almost always to interpret the internal relations and connection mechanisms between the structures and functions of EI. Research on the design of EI, EI-related policy, the participation of EI in environmental governance, and the adaptability of EI, are becoming popular in recent years and promoted the development of EI-related research. EI has developed into a complex system performing multiple service functions such as ecological and socio-cultural functions.

At present, the degradation of urban ecosystem services caused by reckless urbanization and industrialization and the increasing pressure on resources and the environment caused by the change of urban land use affects and restricts the sustainable development of urban areas. As an important concept for guiding urban construction and green space planning, EI embodies planners' deep understanding of urban land use, which is key for creating livable cities and eco-cities. In the future, during the construction of ecological infrastructure, the following points should be noted: first, we propose the establishment of specialized administrative departments of EI to avoid ambiguity and overlap in management; secondly, we propose regarding EI as the framework for land restoration, protection, and development to reflect the foresight of urban planning; thirdly, we propose the construction of ecological networks to emphasize the connectivity of ecosystems; fourthly, we propose the "government-led and multi-stakeholder" development pattern to coordinate the contradiction between ecological protection and serving social masses; lastly, we propose ensuring smooth implementation of EI through legislation.

**Author Contributions:** S.S. drafted the manuscript and performed the article. S.Z. designed the overall framework of this study and made valuable comments and suggestions on the writing and revision. Y.J. collected data. All authors have read and approved this article. All authors have read and agreed to the published version of the manuscript.

**Acknowledgments:** This work was supported by the Strategic Priority Research Program (A) of the Chinese Academy of Sciences (XDA23030402) and the Key Project of National Natural Science Foundation of China (71533003).

**Conflicts of Interest:** The authors declare no conflict of interest.

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
