# Peer review of "Research on Ecological Infrastructure from 1990 to 2018: A Bibliometric Analysis"

_sustainability, doi:10.3390/su12062304_

Round 1

Reviewer 1 Report

The authors made an appreciable attempt to track down the emergence of the "Ecological Infrastructure" into the mainstream scientific literature. Whilst the attempt is good, I found that they need to go through extensive revision work .

  1. First, I am not convinced of the concept of "Ecological Infrastructure". Because, all the authors did is repackaging the emergence of ecological management as the "ecological infrastructure". There are already plenty of researchers available on how ecological movement, governance, approaches become mainstream through systematic reviews. What is authors contributing except coining the term "ecological infrastructure"?. 
  2. From section 3.6.1m, the manuscript lost its focus. Previously, it was based on their bibliometric search results and somewhat interesting and the four sections of 3.6. are completely out of relevance and repetition of the same discussion available in hundreds of scientific papers on ecological, climate and environmental management. I suggest to drop those sections. 
  3. Some minor comments: 
    •     LINE 34: What is eco-environment? ecological or environmental problems should be enough 
    • lINE 38:  nature conservation instead of natural
    • L 41: Please use full meaning of GIS
    • L 104:  Using only one search term is a key weakness of the method which needs to be mentioned
    • L 108-110: How 2359 were selected should be mentioned
    • L 115-117: Should be part of the methodology
    • Fig 5: Timeline started from 1996. Does it mean that was there no publication before?
    • Table 2: Authors should present the titles of the articles. it will help readers to understand what types of literature is being considered
  4. L 250-251: I do not think these papers really contributing to EI. These are general environmental/ ecological papers with a global focus of challenging the social order.
  5. After reading the section 3.4, I am not convinced about the usability of the term "Ecological infrastructure". There are general environmental papers and authors are re-branding it as "EI". For example, if paper four is an EI paper, then the whole IPCC reports should be too. The authors need to define the breadth of EI with the thematic and spatial boundary.
  6. The whole section 3.5 should be significantly reduced. The current length is unnecessary and the authors can briefly discuss the hot-spots in more concise formate. I suggest reduce half of its words, at least
  7. This is a good graphic but the quality has to be better readable. Its not very clear
  8. Section 3.6.1: Not sure what the authors tried to address. it better to drop the section completely.

    From section 3.6.1, the manuscript has lost its direction. Previously, it has discussed the emergence of environmental management research (i would say so instead of EI) and then the next three sections until 3.6.4 is not compatible and not so much based on their results.

    I suggest authors should focus more on the emergence of broader ecological/environmental research related to global sustainability.

Author Response

Dear reviewer:

Thank you for reviewing the manuscript entitled “Research on ecological infrastructure from 1990 to 2018: A bibliometric analysis” despite your busy schedules. We have read your comments and suggestions carefully, we have made extensive modifications to the manuscript, and supplemented extra data to make the results convincing. Point-by-point responses to you are organized and listed in the following document. We expect the revised manuscript to be published in sustainability.

Yours,

Shoukai Sun

RE: sustainability-725408

We would like to thank you for the comments and suggestions for our manuscript entitled “Research on ecological infrastructure from 1990 to 2018: A bibliometric analysis”. The following are our responses to your comments.

Explanation: Major modified or added parts were highlighted in yellow in the revised manuscript.

Q1: In this manuscript, the authors should clarify the concept of Ecological Infrastructure (EI), and explain the differences between EI and other urban ecological spaces.

Response: Thank you for asking this question. Based on your suggestion, we clarified the historical context (line 44-73) and definition of EI (line 92-97) to reveal the contribution of EI-related scholars to environmental problems.

Q2: From section 3.6.1, the manuscript loses its focus. The contents are out of relevance, and repetition of the previous discussion available. The authors should drop those sections.

Response: Thank you for the suggestion. The time-zone view illustrates the evolutionary trend and academic relevance of EI-related research field. We explained the EI-related research frontiers in detail, which reveals the future development trend of EI-related research. Based on the above views, we think that this part of the manuscript should be retained, but will be further modified and improved.

Q3: The author should mention the limitations of using only search term.

Response: Thank you for the suggestion. we explained the advantages and limitations of select “Ecological Infrastructure” as the only search term (line 146-152)

Q4: The timeline starts from 1996 (Fig. 5), and the author should determine if any publications has been published before 1996.

Response: Thank you for pointing this out. Figure 5 shows the annual distribution characteristics of the top 10 most productive academic journals from 1990 to 2018, but the earliest EI-related publication of these journals was published in 1996.

Q5: The literature, “A framework to analyze the robustness of social-ecological systems from an institutional perspective” and “Global freshwater resources: soft-path solutions for the 21st century”, do not contribute to EI-related research. The author should elaborate on the relevance of the content of these papers to EI.

Response: Thank you for asking this question. The manuscript further elaborates on the relevance of EI to two literature, “A framework to analyze the robustness of social-ecological systems from an institutional perspective” and “Global freshwater resources: soft-path solutions for the 21st century”. (line 303-312, 312-316).

Q6: The author should make Figure 10 more readable.

Response: Thank you for asking this question. The time-zone view illustrated the evolutionary trend of EI-related research field (Fig.10), in order to reveal EI-related research frontiers, we added the keywords burst detection to show the keywords that have broken out recently (Fig. 11).

Reviewer 2 Report

Introduction

Research Methods and Data Source

The author has intelligibly arranged the body of "introduction" and  "Research Methods and Data Source". I think even if readers who may be not expertise in his field, they can understand easily.

Line 133 to
The author should explain the selection criteria or reason for "primary journals" you mentioned. Most of the researchers who are belonging to the field such as landscape or ecology etc may already know that the journals you mentioned are positioned as major parts of the fields. However, in order to indicate in the paper, you should present proof such as a statistical source or citation.

Figure4
The author skipped the description of "others" although the "others " composed 83.9% on your result. If the "others" should have been excluded from this paper, you should clarify why.

Line 176
Referring the figure 6, Canada also seems to be the same grade as Australia and the UK. Why did you exclude Canada in the sentence?

Line 179
There is no evidence or cannot find it in this paper that those countries (New Zealand, Denmark, South Africa, France, Belgium, Britain, Switzerland, and Japan) have an important communication hub for cooperative networks. I think this sentence, which is not linked to the evidence data, does not fit in this context. If the author wants to stick to this, the author should add supplement data.

Line 216
Looking at the Table1, The U.S and China have shown outstanding research quantity in the past decade (2010-2018). Australia is in the second category (2000-2009).

Line 217
It may be interpreted very subjectively in this paper to mention that the Chinese research institution, Chinese Academy of Sciences, has formed a team related to EI. I think this may be because the author belongs to this organization. Therefore, the author should provide public quotations such as reports.

3.5 Research Hotspot
The author extracted keywords form 2385 papers and categorized them according to resources(table3). However, the result is not an era trend in research from 1990 to 2018 but integrated keywords for about 30 years. Therefore, if possible, I'd like to suggest to extract and describe keywords by a period such as table1, in order to reinforce the purpose of this paper. 

Line 297 to
What is the standard of the selected "Climate change, Ecosystem service, Biodiversity, Green infrastructure, Conservation" among the more than 25 keywords you categorized?

Conclusion
First, this paper is weak in interpretation and contemplation compared to the valuable result data. There is also a lack of connectivity to the research questions mentioned by the author in the introduction. In the conclusion part of this paper, there are brief summaries of results without in-depth consideration.

In addition, the limitations, as well as the direction toward subsequent studies, should be presented in accordance with the considerations. This paper is a valuable paper, but the source used in this study is not a body of the paper, but a title, abstract, and keyword, so the author should point out the limitations. 

The content of Line 232 is regarded to be an issue that can be considered and discussed fully in the conclusion part. However, the content was not considered in the part at all.

When it comes to considering the whole body of this paper, the content of Line 466 is awkward. Checking the result data presented in this paper, this is because CAS has the most abundant quantities of publications in the world, but is not conspicuous about the number of quotes.

Author Response

Dear reviewer:

Thank you for your constructive comments and suggestions concerning the manuscript entitled “Research on ecological infrastructure from 1990 to 2018: A bibliometric analysis”. These comments and suggestions are valuable and helpful for improving our manuscript. We have seriously discussed about the requirements of reviewer, and tried best to modify our manuscript. Point-by-point responses to you are organized and listed in the following document. We expect the revised manuscript to be published in sustainability.

Yours,

Shoukai Sun

RE: sustainability-725408

We would like to thank you for the comments and suggestions for our manuscript entitled “Research on ecological infrastructure from 1990 to 2018: A bibliometric analysis”. The following are our responses to your comments.

Explanation: Major modified or added parts were highlighted in yellow in the revised manuscript.

Q1: In Section 3.2, the author should explain the selection criteria or reason for "primary journals" you mentioned.

Response: Thank you for the suggestion. In Section 3.2, based on your suggestions, we selected to replace "primary journals" with "major journals" to express the author's intention directly, and clarified the selection criteria for "major journals" (line 183-185).

Q2: In Figure 4, the author clarified the description of "others".

Response: Thank you for pointing this out. Based on Figure 4, we further analyzed the distribution of literature in different journals (line 186-187).

Q3: Referring Figure 6, the author should consider rewriting Section 3.3.1 to reveal the publication distribution of countries.

Response: Thank you for asking this question. We added Table 1 to the manuscript to help interpret Figure 6. We focused on countries with outstanding contributions and countries with high betweenness centrality (line 228-234).

Q4: The author should carefully confirm the accuracy of line 216 of the original manuscript.

Response: Thank you for pointing this out. Table 1 only listed the top 10 most productive institutions that participated in EI-related research in the three periods of 1990–1999, 2000–2009, and 2010–2018, and failing to reveal the participation frequencies of different countries. In line 271 of the revised manuscript, the conclusion “China and Australia have made remarkable progress in EI-related research in the past decade”, is made by comparing Figure 8 with Figure 9. During this period, the number of EI-related institutions and publications in both countries increased significantly.

Q5: The author should carefully confirm the accuracy of line 217 of the original manuscript.

Response: Thank you for asking this question. We rewrote the line 217 of the original manuscript (line 272-274).

Q6: When conducting the analysis of research hotspots, the author chose to extract keywords directly from 2385 papers, without splitting into three intervals of 1990-1999, 2000-2009, 2010-2018. The author should arrange keywords according to the format of Table 1.

Response: Thank you for asking this question. Before the first submission, we arranged keywords according to three time-intervals, but the result was not satisfactory. The reason for analysis is that from 1990 to 1999, there were only 87 EI-related publications, which could not provide sufficient literature resources for extracting keywords.

Q7: The author should expound the selection criteria for keywords such as Climate change, Ecosystem services, Biodiversity, Green infrastructure, Conservation.

Response: Thank you for the suggestion. We added to present the selection criteria for keywords (line 329-334).

Q8: The author should provide contents about the relationships between EI-related research questions mentioned in the introduction, and make in-depth consideration in the conclusion part of this manuscript.

Response: Thank you for pointing this out. We briefly covered the EI-related research history in the introduction (line 44-97), and make in-depth consideration in the conclusion (line 519-531).

Reviewer 3 Report

The manuscript deals with the so called ecological infrastucture (EI), as the composite system on which sustainable development of urban depende. The scientometric approach to describe the current situation of EI-related research has been applied. In the introduction the information of the current status seems to widley describe. The manuscript contains enough deep discussion and very good presented results.

Author Response

Dear reviewer:

Thank you for reviewing the manuscript entitled “Research on ecological infrastructure from 1990 to 2018: A bibliometric analysis” despite your busy schedules. We were pleased to receive the favorable comments of reviewer, your affirmation and recognition greatly inspires our confidence. In order to express the topic and purpose of the manuscript more accurately, we actively revised the details and content of the manuscript. We hope that you will review and correct again.

Yours,

Shoukai Sun

“Research on ecological infrastructure from 1990 to 2018: A bibliometric analysis”. The following are our responses to your comments.

Explanation: Major modified or added parts were highlighted in yellow in the revised manuscript.

Thank you for reviewing the manuscript despite your busy schedules. Your affirmation and recognition greatly inspired our confidence. Based on valuable comments from reviewers, in order to express the purpose of the manuscript more accurately, we actively revised the details of manuscript and supplied the descriptive contents. We hope you take the time to review the modified manuscript and provide advice.

Round 2

Reviewer 2 Report

I agree with the publication of this paper.